# Autophagy, Unfolded Protein Response, and Neuropilin-1 Cross-Talk in SARS-CoV-2 Infection: What Can Be Learned from Other Coronaviruses

**DOI:** 10.3390/ijms22115992

**Published:** 2021-06-01

**Authors:** Morvarid Siri, Sanaz Dastghaib, Mozhdeh Zamani, Nasim Rahmani-Kukia, Kiarash Roustai Geraylow, Shima Fakher, Fatemeh Keshvarzi, Parvaneh Mehrbod, Mazaher Ahmadi, Pooneh Mokarram, Kevin M. Coombs, Saeid Ghavami

**Affiliations:** 1Autophagy Research Center, Shiraz University of Medical Sciences, Shiraz 7134845794, Iran; Morvarid.siri@gmail.com (M.S.); mozhdeh.zamani63@gmail.com (M.Z.); 2Endocrinology and Metabolism Research Center, Shiraz University of Medical Sciences, Shiraz 7193635899, Iran; suny.respina@gmail.com; 3Department of Biochemistry, Shiraz University of Medical Sciences, Shiraz 7134845794, Iran; nasimrahmani86@gmail.com (N.R.-K.); shima1979f@yahoo.com (S.F.); fatmehkeshavarzi@gmail.com (F.K.); 4Student Research Committee, Semnan University of Medical Sciences, Semnan 3514799422, Iran; kiarashgrl@yahoo.com; 5Influenza and Respiratory Viruses Department, Pasteur Institute of Iran, Tehran 1316943551, Iran; mehrbode@yahoo.com; 6Faculty of Chemistry, Bu-Ali Sina University, Hamedan 6517838695, Iran; ahmadi.mazaher@yahoo.com; 7Department of Medical Microbiology and Infectious Diseases, Rady Faculty of Health Sciences, Max Rady College of Medicine, University of Manitoba, Winnipeg, MB R3E 0J9, Canada; kevin.coombs@umanitoba.ca; 8Department of Human Anatomy and Cell Science, Rady Faculty of Health Sciences, Max Rady College of Medicine, University of Manitoba, Winnipeg, MB R3E 0J9, Canada; 9Faculty of Medicine, Katowice School of Technology, 40-555 Katowice, Poland

**Keywords:** autophagy, COVID-19, endoplasmic reticulum stress, SARS-CoV-2, unfolded protein response

## Abstract

The COVID-19 pandemic is caused by the 2019–nCoV/SARS-CoV-2 virus. This severe acute respiratory syndrome is currently a global health emergency and needs much effort to generate an urgent practical treatment to reduce COVID-19 complications and mortality in humans. Viral infection activates various cellular responses in infected cells, including cellular stress responses such as unfolded protein response (UPR) and autophagy, following the inhibition of mTOR. Both UPR and autophagy mechanisms are involved in cellular and tissue homeostasis, apoptosis, innate immunity modulation, and clearance of pathogens such as viral particles. However, during an evolutionary arms race, viruses gain the ability to subvert autophagy and UPR for their benefit. SARS-CoV-2 can enter host cells through binding to cell surface receptors, including angiotensin-converting enzyme 2 (ACE2) and neuropilin-1 (NRP1). ACE2 blockage increases autophagy through mTOR inhibition, leading to gastrointestinal complications during SARS-CoV-2 virus infection. NRP1 is also regulated by the mTOR pathway. An increased NRP1 can enhance the susceptibility of immune system dendritic cells (DCs) to SARS-CoV-2 and induce cytokine storm, which is related to high COVID-19 mortality. Therefore, signaling pathways such as mTOR, UPR, and autophagy may be potential therapeutic targets for COVID-19. Hence, extensive investigations are required to confirm these potentials. Since there is currently no specific treatment for COVID-19 infection, we sought to review and discuss the important roles of autophagy, UPR, and mTOR mechanisms in the regulation of cellular responses to coronavirus infection to help identify new antiviral modalities against SARS-CoV-2 virus.

## 1. Introduction

Viruses, the tiniest causes of infection, with a diameter of 20–300 nm, possess either RNA or DNA as their genetic material. The viral genome is surrounded by a protein capsule called the capsid, which, for some viruses, might in turn be enclosed in a lipid membrane. The single viral particle unit is called a virion. Since viruses are only able to replicate in living cells, they are obligate intracellular parasites. Viruses use their nucleic acid information to take advantage of the infected host cell’s biosynthetic machinery to synthesize 100–1000 s of progeny virus particles in each infected cell. During virus replication, many nucleic acid copies are produced, which are enclosed in newly synthesized and assembled viral capsid proteins to serve as a protective coating while the virion is in the extracellular environment. The adverse effect of viral infection on the host cell varies from slight or no effect, to cell injury or death. RNA viruses are more susceptible to mutations than DNA viruses, probably because of the lack of proofreading during RNA replication. This process results in the variation seen in many RNA viruses, allowing them to evolve more rapidly and escape host cell anti-viral strategies including immune surveillance. Many examples of RNA virus-induced human infections include the common cold, influenza, hepatitis C and E, Ebola, polio, measles, and severe acute respiratory syndrome (SARS) [1].

Coronaviruses (CoVs) are enveloped particles with a single-stranded positive-sense RNA genome of approximately 26–32 kilobases (kb) [2]. They belong to two subfamilies, named *Coronavirinae* and *Torovirinae,* belonging to the *Coronaviridae* family and the *Nidovirales* order. Based on the International Committee for Taxonomy of Viruses (ICTV), the four major genera of the *Coronavirinae* subfamily include α, β, γ, and δ-CoVs [3]. The α and β-CoVs are responsible for human infections and are derived from bats and rodents, while the γ and δ-CoVs evolved from avian species [4]. The α-CoVs (including HCoV-NL63 and HCoV-229E) and the β-CoVs (including HCoV-OC43 and HCoV-HKU1) are responsible for relatively minor infectious diseases such as the common cold, mild upper respiratory tract infections in immunodeficient hosts, and occasionally, virulent infections in the elderly, young children, and infants [2,4,5]. Severe respiratory diseases are caused by highly pathogenic β-CoVs, including SARS-1, responsible for the 2003 pandemic and prevalence of severe respiratory infection, and the Middle East Respiratory Syndrome (MERS-CoV) outbreak in 2012 in Middle Eastern countries [2,4]. Four structural proteins are encoded by all CoVs, including spike (S), membrane (M), and envelope (E) as membrane-associated proteins, as well as nucleocapsid (N) protein [6]. The S protein attaches to the host cell and mediates membrane fusion. The S protein also helps promote the attachment of infected cells to nearby non-infected cells, thus increasing viral transmission and spread. The N protein has a vital role during assembly of new virus particles. It attaches to the viral RNA genome to generate the ribonucleoprotein nucleocapsid core. Since it is the most highly expressed protein in the virus, it serves as an evaluating marker in the early stages of viral infection. The E protein is a small integral membrane protein that includes a short N-terminal domain followed by a large hydrophobic transmembrane domain and a less hydrophobic C-terminal domain [7]. Coronavirus replication follows the attachment of the S protein to host cell receptors. These human cell receptors used by coronaviruses include aminopeptidase N for HCoV-229E, 9-O-acetylated sialic acid for HCoV-OC43 and HCoV-HKU1, angiotensin-converting enzyme 2 (ACE2) for HCoV-NL63, and dipeptidyl peptidase 4 (DPP4) for MERS-CoV [8].

On 31 December 2019, in Wuhan, China, a novel coronavirus was identified as the source of a new severe acute respiratory disease. This disease, known as coronavirus disease-19 (COVID-19), is caused by a new coronavirus named SARS-CoV-2 which rapidly spread worldwide [9]. The genomic material of SARS-CoV-2 is a messenger RNA (mRNA), which acts as a template for protein synthesis and for genomic replication, which together lead to the formation of new virus particles [10]. The first two-thirds of the SARS-CoV-2 RNA is responsible for synthesizing the replicase enzymes, and the other one-third encodes other viral proteins such as S, E, M, and N structural proteins and a large number of non-structural proteins. According to structural model analyses, the binding affinity of SARS-CoV-2 S protein to the ACE2 receptor is about 10- to 20-fold higher than the required threshold for virus infection compared to SARS-1 S protein [11]. So far, the focus has been entirely on ACE2 as a receptor, while recent studies have described the role of a second type of cell surface receptor, called neuropilin-1 (NRP-1), in the entrance of the virus into host cells and the virus’ subsequent replication. SARS-CoV-2 is more transmissible and persistent compared to SARS-1, perhaps because of its ability to spread rapidly through active pharyngeal viral shedding, the presence of a polybasic furin-type cleavage site at the S1–S2 junction in its spike protein, which is absent in SARS-1, and the use of both ACE2 and NRP-1 receptors to infect cells [12].

Furin inhibitors decanoyl-RVKR-chloromethylketone (CMK) and naphthofluorescein decrease virus production and have antiviral effects on SARS-CoV-2-infected cells by abolishing cleavage and syncytium formation. CMK blocks the entry of viruses and naphthofluorescein suppresses viral RNA transcription. Therefore, they could be promising antiviral agents to reduce virus spread and treatment of SARS-CoV-2 infection [13].

Endoplasmic reticulum (ER) stress is involved in maintaining cell homeostasis, lipid synthesis, protein folding, translocation, and post-translational modifications. Various stresses, including hypoxia, starvation, pH changes, calcium depletion, and viral infections, can interfere with the ER environment [14]. Viral infection and toxicity may cause ER stress through the use of the ER membrane, aggregation of misfolded proteins, calcium imbalances due to the formation of viroporins, and destruction of the ER membrane while the virions are embedding [15]. β-CoV replication in the cytoplasm depends on the virus’ life cycle in the ER. When SARS-CoV-2 replicates, it produces a large number of viral proteins. The S, M, and some other proteins are highly modified and integrated as membrane proteins, which increases the chance of unfolded coronavirus protein accumulation in the lumen of the ER, because of their limited folding potential. The accumulation of unfolded or misfolded proteins leads to the activation of the unfolded protein response (UPR) pathway, resulting in a cascade of signaling pathways to the cell’s nucleus to restore ER homeostasis. ER stress is a situation in which the cell needs to exceed the strength of the ER to cope with stress [16]. The UPR is an adaptive defense signaling pathway affected by protein kinase R (PKR)-like ER kinase (PERK), inositol-requiring enzymes (IRE-1), and activated transcription factor 6 (ATF-6) (Figure 1). The aforementioned sensors are located on the ER and act as key arms for triggering the UPR pathway in health and disease [17,18]. UPR activation reduces the synthesis of misfolded proteins by inhibiting the synthesis of new proteins and increasing the ability of protein folding. Under stress, glucose-regulated protein 78 (GRP78) is overexpressed to trigger UPR responses and regulate disordered proteins [17]. GRP78 is a heat shock protein (HSP) located in the ER and responsible for several important intracellular processes, such as facilitating the transfer, completion, and accumulation of newly produced proteins in the cell, and preventing the accumulation of misfolded proteins [19]. Generally, as a sensor and regulator, it can recognize ER stress and can serve as the main trigger for three UPR sensors or arms [20,21].

The PERK arm of the UPR, as a serine/threonine transmembrane protein, is located on the ER membrane. Under normal cell conditions, it is a passive monomer that binds to GRP78. In contrast, GRP78 dissociates and PERK becomes oligomerized and activated during stress [22]. Activated PERK has two specific functions, activating eIF2α-ATF4, which inhibits protein synthesis as well as activating CHOP protein (C/EBP homologous protein), which activates cell death receptor 5 (DR5), and growth arrest- and DNA damage-inducible gene 153 (GADD153) which leads to cell growth arrest. Therefore, this process changes cell fate from cyto-protective to pre-apoptotic cell death and initiates the caspase cascade [23]. In humans, there are two paralogues of IRE1 (IRE1α and β), located on the ER as another UPR sensor with both kinase and endoribonuclease activity in the cytosolic domain. Among the UPR arms, IRE1α is a key sensor for regulating the cell’s fate. GRP78 dissociation, caused by ER stress, triggers IRE1 oligomerization and activates its cytosolic domains, leading to atypical splicing of X-box binding protein (XBP-1) mRNA as a transcription factor [24]. Activated XBP-1s can regulate the expression of several genes that are involved in the proper folding of proteins, trafficking, and the secretion and restoration of ER capacity. In addition, XBP-1s reduces ER proteins by affecting nuclear factor-c-Jun NH2-terminal kinase (JNK), NFκB, and inositol-requiring enzyme decay RIDD (IRE1-dependent decay), and reduces the expression of certain proteins, mRNA, and microRNAs [25,26]. ATF6 (activating transcription factor 6) is a type 2 transmembrane protein that contains the N-terminal cytoplasmic domain with a DNA-binding motif. ATF6 needs to transport to the Golgi compartment after dissociation from GRP78 to become activated by regulated intra-membrane proteolysis (RIP) with site-1 and site-2 (SP) protease. ATF6 can induce the expression of ERAD (ER-associated degradation) components, ER chaperones, quality control, and protein folding machinery, and various other transcription factors such as the components of the nuclear transcription factor Y (NF-Y) complex and serum response factor as well as XBP1 and CHOP to enhance UPR signaling [27,28].

Autophagy is a highly conserved pathway in which a double membrane layer surrounds defective organelles or intracellular components, such as accumulated proteins, to form an autophagosome, which then binds to a lysosome to form an autolysosome, which degrades the components (Figure 2). The autophagy pathway is controlled by proteins translated from autophagy-related-genes (ATGs) in sequential steps. The initiation stage is managed by the ULK1/ATG1 complex, which is located downstream of the mechanistic target of the rapamycin complex 1 (mTORC1). The expansion, nucleation, or elongation step is controlled by the ATG14-Beclin1-hVPS34/class III phosphatidylinositol 3-kinases (PI3K) complex and two ubiquitin-like conjugation systems (ATG5-ATG12 and LC3/ATG8). The final stage of autophagy is the fusion, degradation, and maturation stage, which includes the fusion of the autophagosome with the lysosome to form an autolysosome to degrade the contents [29].

The UPR is a key pathway of ER stress, which stimulates autophagy through its three arms; PERK, IRE1α, and ATF6α (Figure 1) [30]. As the main regulatory step of this pathway, IRE1α mediates the phosphorylation of the stress-associated protein kinase, MAPK8 (mitogen-activated protein kinases 8). Activation of MAPK8 induces autophagy and apoptosis. The interaction of JNK with MAPK8 triggers autophagy downstream mediators in a direct and indirect manner [31]. In addition to JNK, IRE1α induces autophagy through tumor necrosis factor (TNF) receptor-related factor 2 (TRAF2) [32]. The IRE1α-XBP1s axis is involved in the initiation and maturation of autophagy by indirect regulation of B-cell lymphoma 2 (Bcl-2) expression and microtubule-associated protein light chain I (LC3- I) lipidation to LC3-II [33]. XBP1s also upregulates the transcriptional expression of the *Beclin1* gene through direct binding to the −537 and −755 regions of its promoter [34]. The role of the PERK arm in ER stress-mediated autophagy was first described by Kouroku et al. [35]. Proteasome activities are decreased after the aggregation of the polyglutamine (72Q) protein in the cytosol, which results in PERK-mediated autophagy [35]. PERK also activates the transcription of the LC3 and ATG5 proteins during hypoxia response through the induction of transcription factors ATF4, C/EBP homologous protein (CHOP), and DNA damage inducible transcript 3 (DDIT3). Reduction of IkBα translation and NF-κB activation are also mediated by PERK, which promotes autophagy induction [31]. PERK is responsible for eIF2α phosphorylation at the serine 51 residue and ATF4 activation, which leads to the expression of autophagy-related genes such as *ATG3* and *ATG12* [31]. The third arm of the UPR is an ATF6α transcription regulator, which initiates autophagy by enhancing the expression of heat shock 70 kDa protein 5 (HSPA5) and inhibiting the expression and activation of protein kinase B of AKT1/AKT [36].

Both autophagy and UPR participate in the pathogenesis of viral infections [37,38]. UPR and autophagy might function dependently, or the activation of one may affect the other one. The precise roles of UPR and autophagy in viral infections in humans remains to be investigated. Many experimental studies have shown that various viruses use these mechanisms to escape the host immune system or even take advantage of them for their benefit, which led to much current attention in the area of antiviral study [39]. Autophagy and UPR are tightly regulated by specific molecules that can facilitate virus replication. Hence, the role of these two mechanisms needs more precise investigations in viral infections. Since SARS-CoV-2 infection has no specific medication yet, manipulation of autophagy and UPR might be a novel approach for combating SARS-CoV-2. In this review, we provide evidence linking UPR and autophagy to coronaviruses and discuss whether such links may provide actionable targets for therapeutic interventions.

## 2. Coronavirus-Induced ER Stress Response Activates the UPR

Coronavirus enters host cells by binding to cell surface receptors and produces viral progeny particles by replicating its genome and translating its proteins [40]. These events elicit ER stress and ultimately activate ER stress sensors to promote UPR responses for re-establishing homeostasis [37,41].

In a viral infection, the unfolded proteins bind to GRP78 to release ER stress transducers such as PERK, ATF6, and IRE1. Following dimerization, PERK is activated by autophosphorylation. Then, the active PERK phosphorylates eIF2α, thereby attenuating general translation and induces GADD34 and CHOP synthesis. When ATF6 is released from GRP78, it translocates to the Golgi apparatus, where it is cut into a truncated form that stimulates the expression of molecular chaperone genes. Once IRE1 is released from GRP78, the activated IRE1 promotes XBP1 mRNA splicing. It is worth noting that virus-encoded proteins interfere with these signaling pathways, thus leading to altered viral replication rates [42].

It is interesting that virus-triggered ER stress differentially activates the PERK, IRE1, and ATF6 branches of the UPR. Host cells prefer to induce the PERK pathway in response to viral infection probably because the attenuation of global translation, mediated by PERK, can effectively limit viral replication by inhibiting viral and cellular protein synthesis, which is critical for the viral life cycle [43]. The IRE1-XBP1 pathway may increase virus replication by increasing ER protein folding and membrane biosynthetic capacity [43]. However, activation of the ATF6 pathway in cells infected by coronavirus increases chaperone protein levels to overcome ER stress [41]. The ER stress responses produced by the three sensor proteins are differentially induced by various coronaviruses [44].

Recently, studies have shown that 5 h after murine coronavirus, strain A59, infection, p-eIF2α and ATF4 can be elevated without any GADD34 and CHOP induction [45]. XBP-1 caused by the activation of the IRE1 pathway has not been observed in SARS-1 infection [46]. Moreover, no sign of ATF6 pathway activation has been detected in coronavirus infection. In murine hepatitis virus (MHV; a coronavirus), strain A59 infection, the active form of ATF6 can still be observed in the initial stage but is significantly reduced in the late stage [47]. This cleaved form of ATF6 has not yet been found in SARS-1 infected cells [48]. The SARS-1 8ab protein is a group-specific accessory protein. Due to the deletion of 29 nucleotides in the ORF 8ab region, the protein will be lost during virus transmission from animals to humans. The 8ab protein is attached to the luminal surface of the ER membrane. The 8ab protein can elevate endogenous ER-resident chaperone synthesis by activating the transcription factor ATF6 with no effect on the induction of CHOP and XBP1 splicing associated with UPR. These findings indicate that 8ab can promote protein folding and processing by activating ATF6, thereby regulating the UPR. The loss of 8ab in SARS-1 through virus evolution in animals may play an important role in its pathogenicity (Figure 3) [49]. In addition, viral E protein can form Ca^2+^ channels on the ERGIC/Golgi membrane, and the resultant imbalance in Ca^2+^ homeostasis can cause ER stress and cell death. The avian infectious bronchitis (IBV) coronavirus’ E protein can also activate ER stress by forming Ca^2+^ channels in the ER of host cells [50]. 

IBV infection positively affects the eIF2-ATF4-GADD153 pathway to regulate stress-induced apoptosis. Increased rates of PERK and eIF2 phosphorylation can be detected during the initial stage of IBV infection [51].

In conclusion, three UPR branches cooperate as a network and are not functionally independent. Recent studies have shown that UPR can interfere with other cell signal transduction pathways, including the MAP kinase pathway [52], innate immune response [53], and autophagy [54]. Similar to other coronavirus infections, SARS-CoV-2 causes oxidative and ER stress, which triggers cellular response pathways, mainly the PERK and IRE1 branches of the UPR. PERK can be induced by SARS-CoV-2 infection and may be mediated by the S and 3a proteins. The IRE-1 branch is also affected by SARS-CoV-2. Therefore, significant XBP1 splicing and increased apoptosis levels have been detected in infected cells with recombinant SARS-CoV-2 containing the missing E protein. As far as ATF6 is concerned, SARS-CoV-2-induced induction in these branches has not been reported [55]. Therefore, further research on UPR affected by the coronaviruses may also help identify new strategies for antiviral treatments and led to the development of more effective anti-coronavirus vaccines.

### 2.1. GRP78 Facilitates Surface Coronavirus Attachment

GRP78, also known as BiP or HSPA5, was considered an ER chaperone, but recent studies have investigated the role of GRP78 as an attachment protein on the surface of host cells for receptor-mediated entry of two Beta-coronaviruses, MERS-CoV and bCoV-HKU9. New studies have revealed that GRP78 can only increase MERS-CoV entry into permissive cells and make them susceptible to infection by improving virus attachment. GRP78 is highly conserved and interacts with bat coronavirus spike protein HKU9 and increases virus entry into host cells [56]. A recent in silico study showed that similar to MERS-CoV binding, GRP78 binds to SARS-CoV-2 spike protein. The inhibition of this attachment by drugs may be an efficient approach in reducing COVID-19 infection [57].

### 2.2. Effect of Coronavirus S and E Proteins on the UPR

The process of entering the host cell is an important determinant of virus infectivity and pathogenesis. To enter host cells, coronaviruses must first bind to cell surface receptors. Virus entry is thought to involve membrane-membrane fusion. This may occur at the cell surface or during acidification of the endosome that takes up the virus particle, although there are some questions about how endosomal acidification studies are conducted and interpreted. The spike protein anchored on the surface of the virus mediates the entry of the coronavirus [50]. The coronavirus spike protein can use many cell surface molecules for attachment and entry; therefore, it has recently been predicted that the SARS-CoV-2 spike protein and Pep42 protein (-peptide-42) have four similar regions in the structure and sequence, which can bind to GRP78. In MERS-CoV and bat coronaviruses, GRP78 is attracted to its internalization because it is another target for their spike proteins [56]. Therefore, SARS-CoV-2 can be distinguished by cell surface GRP78 during cell stress, which is one of its advantages [57]. The SARS-1 and HCoV-HKU1 S proteins use different UPR activation domains to play the same regulatory role in the UPR signaling pathway. These two S proteins have three characteristics: First, they are mainly located in the ER. Second, they show similar UPR activation characteristics and can activate the GRP78, GRP94, and CHOP promoters, but not the UPRE enhancer, which is responsive to XBP1- and ATF6-related transcription factors. Third, to activate GRP78 and GRP94 promoters, they need PERK catalytic activity, but not N-linked glycosylation [58]. An in vitro study using selective PERK inhibitors found that reducing translational inhibition increases viral protein levels, but reduces infectious viral titer. Presumably, this is due to increased translation of the host antiviral proteins [59]. Therefore, the activation of UPR by S protein may facilitate further investigation of UPR modulators for the treatment of SARS-CoV-2 and HCoV-HKU1 infections. In this way, the ChAdOx1 nCOV-19 vaccine, developed by the University of Oxford, is composed of the genetic sequence of the SARS-CoV-2 S protein and a non-replicating adenovirus vector, and has been approved [60].

MHV and IBV infections induce the IRE1-XBP1 pathway, which may be related to the accumulation of S protein in the ER lumen. Neither SARS coronavirus infection, nor SARS coronavirus spike protein accumulation, lead to XBP1 splicing, which indicates that the modulation of UPR branches varies among different coronaviruses [51]. The unique strategy of SARS-1 S protein in dealing with ER stress is stimulating PERK, but not IRE1 or ATF6. Consistently, SARS-1 infection did not activate apoptosis in dendritic cells (DC) in the early stages of the disease [61]. This observation is related to the inhibitory effect of E protein on the host cell response. The SARS-1 E protein can also suppress the IRE1-XBP1 pathway and inhibit virus-mediated apoptosis [62]. A recent study observed a significant XBP1 splicing and a higher level of UPR-related gene expression in infected cells with recombinant-E-gene defective SARS-1 (rSARS-CoV-ΔE) compared to the wild type SARS-1 [51]. Therefore, it seems that as long as E protein is present in SARS-1, apoptosis activation following stress is restricted in the infected cells, which is an advantage for virus replication and distribution. Furthermore, another current study showed that loss of E genes from SARS-1 resulted in the up-regulation of heat shock protein-related genes, causing an enhanced recognition of infected cells by immune responses [63], which may even limit proinflammatory responses and pathological effects [48]. Thus, it seems that IRE1 has a crucial role in the activation of apoptosis and immune responses in the SARS family of coronavirus infection. As a result, the E protein seems to occupy a central role in the reduction of UPR and immune responses in SARS-CoV-2 infection. A summary of the studies on ER stress responses in pathogenic coronaviruses is presented in Table 1. The E protein has a key role in the virus life cycle, and E-deficient CoVs may be good candidates for vaccine production and anti-coronaviral agents [64]. 

### 2.3. Antiviral Targets

SARS-CoV-2 can hijack the UPR signaling pathway for its own benefit during the infection process. In this regard, the PERK-eIF2 pathway has gained much attention for its antiviral aspects. An inhibitor of the IRE1 RNAase activity, MKC8866, also represses *XBP1* splicing, resulting in the modulation of innate immunity through the UPR. MKC8866 has been shown to be effective in cancer models by targeting the IRE1/XBP1 pathway [67,68,69]. The PERK pathway can also be inhibited by the GSK-PERK inhibitors in order to modulate innate immunity by the activation of type 1 interferon (IFN) signaling [37]. Therefore, inhibitors of PERK (i.e., the GSK-PERK inhibitor) and IRE1 RNAase activity (MKC8866) are able to adjust the UPR response in infected cells through modifying innate immunity. Therefore, these inhibitors seem promising as possible therapeutics for COVID-19 infection [37].

Moreover, viral proteins such as SARS-CoV-2 proteins that trigger ER stress can be targeted as antiviral therapies. Andrographolide and Melatonin might be useful as antivirals in COVID-19 infection since they can alter ER stress because of their anti-inflammatory, anti-oxidant, anti-pyrogen, and immunomodulatory features. Andrographolide has a variety of biological activities, including immunomodulation and determining the SARS-CoV-2 binding site. Andrographolide antiviral mechanisms include altering the ER stress-mediated UPR pathway and inhibiting the main protease of SARS-CoV-2 [70]. Melatonin stimulates UPR and unlocks autophagy blockage, allowing the formation of autophagolysosomes, which results in complete autophagy and a decrease in viral replication capacity. In addition, Melatonin has been reported to show protective effects against some drugs and toxins, making the combined therapy more effective [55,71]. In short, coronavirus replication leads to ER stress and UPR induction. Therefore, investigating coronavirus-induced UPR can introduce new antiviral therapy targets against SARS-CoV-2-infected cells.

### 2.4. Effect of UPR on the Innate Immune Response 

Innate (also called non-specific) immunity is the first line of defense in the immune system, starting from the first moment of exposure to a virus, and lasting for up to three days post infection. Innate immunity works through a large number of pattern recognition receptors (PRRs), which can detect different evolutionarily conserved molecules on pathogens, called PAMPs (pathogen-related molecular patterns). The TLR (toll-like receptor), RIG-1 (retinoic acid-inducible gene-I-like receptor), and NLR or NOD-like receptors (nucleotide-binding oligomerization domain-like receptor), are located on the cell surface, in endosomal membranes, and in the cytoplasm, and can be used as PRRs to recognize different types of PAMPs [72]. Intracellular pathogens such as viruses stimulate the immune response by activating endosomal and cytoplasmic PRRs. The host cells are exposed to the viral dsRNA (double-strand RNA; a normal intermediate of RNA replication) during viral replication and are identified by RIG-I, endosomal-localized TLR3, MDA-5 (melanoma differentiation-associated protein 5), and interferon-induced sensors such as PKR [73], as shown in Figure 4.

*IRE1:* The RNase L domain of IRE1 detects viral infections by cytoplasmic ribonuclease, which helps degrade viral RNA and activates IFNI (interferon I) and RIG-I/CARDIF pathways of innate immunity [74]. In addition, activated IRE1 can bind and phosphorylate TRAF2 and subsequently induce NF-κB (nuclear factor kappa-light-chain-enhancer of activated B cells), leading to pro-inflammatory cytokine (IL-1, IL-2, TNFα) release (Figure 4) [75]. Moreover, the interaction of IRE1 with the TRAF2 adaptor protein activates JNK [76]. IRE1 is essential for maximizing the induction of TLR and IL-6 production in macrophages, especially under ER stress [77]. The IRE1-XBP1 axis is also necessary for poly I:C (polyinosinic:polycytidylic acid) to promote the production of interferon β (IFN-β) and inflammatory cytokines in dendritic cells [78], and induce IL-8 in cells infected with avian coronavirus. It has also been shown that XBP-1 is essential for increasing the production of IFN-β, IL-6, ISG15 (interferon-stimulated gene 15), TNF-α, and IL-8 in response to the simultaneous activation of PRR and ER stress response. Studies have also revealed that IRE1⁄XBP-1 is of great significance to the vitality, function, and maturity of DC and the differentiation of plasmacytoid dendritic cells (pDC) [79]. Without XBP1, the number of different DC subsets is greatly reduced. In conclusion, these findings indicate that IRE1 plays a synergistic role in innate immunity, acting as a signaling agent or complementary sensor during these infections.

*PERK:* For PERK, three functions are thought to enhance the innate immune response (Figure 4).
PERK can inhibit the translation of IκBα by p-eIF2a, thereby activating NF-κB. Free accumulation of NF-κB causes NF-κB to activate and transfer to the nucleus, which leads to the expression of pro-inflammatory cytokines (IL-6, TNF-α, IL-1) [80];PERK can directly phosphorylate and activate JNK;PERK leads to p38 phosphorylation in the MAPK pathway.

One of the key regulators in innate immunity and ER stress response is GADD34, which is a negative regulator of eIF2α. In cells lacking TLR3 expression, the introduction of viral dsRNA causes GADD34 to express downstream of PERK, which phosphorylates eIF2a and helps the cells restore hemostasis after ER stress [81]. The expression of GADD34 had no significant effect on protein synthesis in dendritic cells, but it was essential for the production of IFN-β and IL-6 as pro-inflammatory cytokines [82]. As a transcription factor activated downstream of PERK, CHOP is transferred to the IL-23 promoter in DC and induces further cytokine production [83]. CHOP is also crucial for IL-6 production in macrophages [84].

*ATF6:* The activation of ATF6 induces NF-κB, but the exact mechanism is unclear. It has been suggested that ATF6 induces NF-κB by encoding XBP-1 and CHOP regulatory genes. In ischemia, ATF6 increases the expression of TNF-α and IL-6, which are macrophage inflammatory cytokines [85]. Defects in the ATF6 pathway can prevent innate immune system activation [86].

UPR-PRR has synergistic effects and participates in the expression and activation of proinflammatory cytokines (IL-6, IL-8, and TNF-α), macrophage cell lines (J774 cells), and enhancement of innate immune responses (Figure 4). The production of cytokines in phagocytic macrophages leads to faster activation of the IRE/XBP-1 sensor in response to TLR stimulation in the TRAF6 and NAPDH oxidase-2 pathways [77]. The UPR response to viral infection is not only a cellular physiological response but also increases synergistic responses to the virus with pattern recognition sensors [87]. UPR stress responses can help host cells overcome virus infection and increase the expression of IL-6, IL-8, and TNF-α, which is observed in infections with SARS-1 S protein via the NF-κB pathway [51]. Therefore, the ER stress response can induce the NF-κB pathway during coronavirus infection [88].

It is crucial to understand the mechanisms involved in the regulation of UPR by immune responses and inflammatory cytokines in viral infection because improving these physiological mechanisms by cytokines may be a novel approach to inhibit the replication and pathogenicity of the virus, and save the lives of patients with chronic inflammation leading to tissue damage.

## 3. Viral Infection and Autophagy

Autophagy may function as a double-edged sword, both as an antiviral pathway and as a pro-viral pathway [89]. Mostly, it is considered the second protective process and a vital response to viral infection in the host cell defense system that manages innate immune system signaling, degradation of pathogens, and adaptive immunity improvement [90,91,92].

When the pathogen is a virus, degradation of viral components is called virophagy, which is a specific type of selective autophagy. In addition to their roles in the main autophagy pathway, the individual autophagy proteins or subdivisions of the core autophagy system can also exhibit antiviral features in the absence of autophagosome maturation. The signals inducing virophagy are PAMPs and danger-associated molecular patterns (DAMPs) [93].

Orvedahl et al., in their study on Sindbis virus, showed that virus nucleocapsids are selectively targeted for autophagic degradation by involving adaptor proteins p62/SQSTM1 and SMURF1 (Smad ubiquitin regulatory factor 1).

The HECT (homologous to E6-AP carboxyl terminus) domain containing E3 ligase in SMURF1 interacts with p62/SQSTM1 [94]. They interact with Sindbis virus nucleocapsid protein in an ubiquitin-independent manner. Therefore, the p62/SQSTM1-SMURF1 pathway demonstrates a novel ubiquitin-independent mechanism of selective autophagy in viral infection [93].

Viruses have developed mechanisms to escape, subvert, exploit or hijack autophagy for their benefit [95,96]. Some viruses escape innate immunity by controlling autophagy [97].

Autophagy-related genes (ATGs) also play important roles in regulating antiviral immune responses, both positively and negatively. They perform as adaptable intracellular transport systems [98].

It was found that ATG13, and to a lesser extent RB1CC1 and ATG7, depletion increase encephalomyocarditis virus (EMCV, a picornavirus) replication. Similarly, overexpression of ATG13 and RB1CC1 reduces virus infection [97].

The absence of autophagy activation results in reactive oxygen species accumulation in dysfunctional mitochondria, amplification of RIG-I-like receptor (RLR) signaling, and cytokine production. Vesicular stomatitis virus (VSV, a rhabdovirus) inhibits autophagy by which the accumulation of ROS in mitochondria enhances RIG-I signaling and inflammasomes [99].

A review in 2009 focused on some ATG involvement in viral infections. Production of type I IFN was enhanced in knocked out (KO) *ATG5* and *ATG7* mouse embryonic fibroblasts (MEFs) in response to VSV infection. The increase in type I IFN production in ATG5 KO MEFs was associated with decreased viral replication. In addition, siRNA-mediated silencing of the *ATG12* gene inhibited intracellular processing of the Epstein Barr virus (EBV) nuclear antigen 1 (EBNA1), and decreased EBNA1-specific CD4+ T cell responses. The siRNA-mediated knockdown of LC3 and *ATG12* in poliovirus-infected cells inhibited the release of infectious virus while only minimally affecting viral replication, suggesting that the primary function of poliovirus’s use of the autophagic machinery was for viral release rather than as a membrane scaffold for RNA replication [89].

In a study on HPV16 early genes by Hanning et al., the anti-autophagic consequence of HPV16 was counteracted by depletion of HPV16 oncogenes which caused autophagy induction. Therefore, they argued against using autophagy inhibitors as a treatment for cervical cancer [100]. 

The murine cytomegalovirus M45 protein binds to the NF-κB essential modulator and limits the over-expression of inflammatory cytokines by blocking TLR and IL-1 receptor-dependent NF-κB activation [101].

It has been reported that influenza virus M2 protein can induce the formation of inflammasomes [102]. A549 human alveolar epithelial cells and MEF cells were used to study autophagy, apoptosis, and the cross-talk between them in IAV infection. A/Puerto Rico/8/34 (PR8) infection of A549 cells induced autophagy and apoptosis simultaneously. However, MEF Bax and Bax/Bak KO cells displayed inhibition of virus-induced cytopathology. Using autophagy-refractory *ATG3* and *ATG5* KO cells, the same results were obtained. These cells did not express LC3β-II and lacked the capacity for autophagy. It was indicated that PR8 infection can induce autophagy and Bax/caspase-dependent apoptosis, simultaneously. It was shown that autophagy plays a role to support PR8 replication by modulating virus-induced apoptosis [103].

Later, it was shown that influenza virus M2 protein can regulate innate immune response. Its proton channel activity is essential for ROS production elevation. Increase in ROS triggers autophagy through the PI3K-AKT-mTOR pathway. The viral M2 protein can also anchor to mitochondria and cause their fusion which results in mitochondrial antiviral signaling (MAVS) protein aggregations which are packed into the autophagosomes and degraded in autolysosomes. The LC3B and ATG5 make a complex with MAVS and attenuate the MAVS-mediated antiviral signaling pathway. However, M2 interacts with MAVS and dissociates these complexes, as well as inhibits autolysosome formation and suppresses the MAVS aggregate degradation. The released MAVS participate in innate immune response, which causes subsequent amplification of inflammatory cytokine signaling by RLR signaling [104].

Hepatitis B and C virus (HBV and HCV) infections have also been shown to activate apoptosis, autophagy and UPR in hepatocytes in different ways, and hijack these processes in their own favor [105]. By studying transgenic mice expressing HCV proteins in the liver, it was found that IFN-β, but not IFN-α, could provoke the autophagic degradation of HCV core and viral NS3/4A proteins. These authors indicated that autophagy could negatively regulate HCV replication in the presence of IFN-β [106]. HCV can negatively regulate the production of type I IFN innate immune responses and promote its replication by autophagy induction and UPR activation [107].

In HBV infection, HBx protein is over-expressed and directly induces autophagy by activation of death-associated protein kinase (DAPK), interaction with c-myc to regulate NF-κB signaling, and mediate Beclin1 expression to induce autophagy [108,109]. HBx also binds to Vps34, the catalytic subunit of phosphatidylinositol 3-kinase class III (PI3KC3) and raises its enzymatic activity and upregulates the autophagic response [110].

In HCV infection, viral NS5A protein upregulates *Beclin1* and induces autophagy. It inhibits autophagy induction by enhancing mTOR [111]. NS4B induces autophagy by directly interacting with Vps34 and Rab5 [112]. NS3, in complex with NS4, interacts with immunity-associated GTPase family M (IRGM), which associates with the autophagy-related proteins ATG5, ATG10, MAP1CLC3 and SH3GLB1 to induce autophagy [113].

HBV and HCV can also induce incomplete autophagic processes by inhibiting lysosomal degradation. In HBV infection, Rab7 is inhibited and lysosomal acidification is impaired by HBx interaction with vacuolar-type H^+^-ATPase (V-ATPase) [114]. In HCV infection, the maturation of autophagosomes is different in each stage of HCV infection. It was found inefficient in the early stage of HCV infection while efficient in the late stage [115,116], which could contribute to differential induction of Rubicon and UVRAG by HCV in different stages, which negatively and positively regulate the maturation of autophagosomes, respectively. Different HCV genotypes have also induced different effects on autophagosome maturation [107].

Overall, these two viruses can induce autophagy to facilitate their replication in host cells. They perform this task with their non-structural proteins, in which direct or indirect association with ULK1/Beclin1/PI3KC3 triggers the UPR to meditate the initiation of autophagy [117]. MERS-CoV also blocks autophagy by decreasing Beclin1 levels and blocking the autophagosome-lysosome fusion to evade degradation and improve membrane availability. S-phase kinase-associated protein 2 (SKP2) ubiquitinates Beclin1, thus promotes its proteasomal degradation. Inhibition (phosphorylation) of SKP2, genetically or pharmacologically, decreases Beclin1 ubiquitination and degradation and enhances autophagic flux and reduces MERS-CoV replication [118].

Recent studies with Japanese encephalitis virus (JEV, an arbovirus) have shown that in cells with impaired autophagy, viral infection induces IFN regulatory factor-3 activation and MAVS aggregation, which are the innate immune response markers of activation [119], suggesting significant inhibition of JEV RNA expression in *ATG5* or *Beclin1* knock-down cells. Inhibition of viral replication was also observed in *ATG7* and *RIG-I* double knock down cells. The N-terminal region of the JEV NS2A contains a sequence which inhibits PKR-induced eIF2α phosphorylation [120].

In murine norovirus (MNV) infection, a series of ATG proteins including ATG5, ATG7, ATG12 and ATG16L1 are involved to disrupt viral replication complexes via an IFN-γ-dependent mechanism [98]. However, it had been shown that MHV replication does not require the autophagy gene *ATG5* in vitro [96].

Younho et al. reviewed norovirus (NoV) through interferon-γ (IFNγ)-inducible GTPases. The ATG5–ATG12–ATG16L1 complex was recruited in the replication complex and restricted virus replication by functioning against autophagosome formation. The increase in LC3-II in Zika virus (ZIKV)-infected placentae and lower titers of ZIKV in ATG16-deficient murine fetuses supported the pro-viral function of autophagy during ZIKV infection [39].

Autophagy induces replication of some single-stranded RNA viruses such as picornaviruses by generating enveloped vesicles containing virions. These pseudo-enveloped virions, generated via an autophagy-like process, contain phosphatidylserine, which increases the penetration rate into adjacent cells. This is how ATG proteins help picornaviruses replicate and penetrate cells [121].

In summary, the exact functions of ATG genes and their proteins and mechanisms to govern autophagy during viral infection still need more detailed evaluation.

### 3.1. Coronavirus and Autophagy

The autophagy–lysosome system plays a central role in the process of infection with different CoVs, including SARS-CoV-2 [122]. In MERS–CoV infection, which is quite pathogenic, virus-induced AKT1 activates the E3-ligase S-phase kinase-associated protein II that inhibits autophagy by targeting the main autophagy initiating protein Beclin1 for proteasomal degradation [123].

In the case of coronaviruses, the membrane-associated papain-like protease PLP2 (PLP2-TM) acts as a novel autophagy-inducing protein via interaction with key auto-phagy mediators such as Beclin and LC3, and regulates antiviral IFN signaling. Moreover, *Beclin1* knockdown can nearly reverse the PLP2-TM’s negative control on innate immunity, resulting in the reduction of coronavirus replication. It can be concluded that coronavirus papain-like protease interacts with *Beclin1* and results in defective autophagy, which leads to the modulation of virus replication and antiviral innate immune system [124].

Since both coronavirus replication and cellular autophagy require ER-derived membranes, the components inhibiting double-membrane vesicle (DMV) generation attenuate coronaviruses replication in vitro [54,118].

In considering various scientific reports on different viruses, different cells and different techniques used in virophagy/autophagy studies, it was suggested that virophagy might also be a beneficial approach for combating SARS-CoV-2 [125].

### 3.2. Antivirals Targeting Autophagy

A previous study on influenza A virus antiviral testing revealed that Baf-A1 in low concentration was an effective inhibitor of IAV replication, without exhibiting toxic effects on A549 cell viability. Baf-A1, as a member of the plecomacrolide subclass of macrolide antibiotics, inhibits vacuolar ATPase and reduces endosome acidification and lysosome number [126].

There are summaries of some antivirals targeting autophagy for HBV and HCV therapy in the review study conducted by Zhang [117]. He mentioned two strategies for targeting HBV-induced autophagy. One way is inhibition of autophagy initiation. In this strategy, miRNA-141 directly targeting the 3-UTR of Sirt1, a sirtuin, plays an important role in autophagy regulation by deacetylating autophagy-related proteins [127]. ATG12 might also serve as a potential target in HBV therapy by inducing antiviral innate immunity and suppressing autophagy [128]. The miRNA-192-3p can reduce HBV propagation in vivo and reduces the levels of HBeAg, HBsAg and DNA/RNA in the blood and livers of mice [109].

The other strategy is enhancement of lysosomal degradation. Epigallocatechin-3-gallate (EGCG), a polyphenol compound, can induce lysosomal acidification and complete autophagic process [129]. PRKAA, a catalytic subunit of AMPK activated in response to oxidative stress induced by HBV, promotes autolysosome-dependent degradation through stimulation of cellular ATP levels which leads to a decrease in HBV replication [130].

Regarding HCV antivirals targeting autophagy, it was shown that *ATG5* silencing can interrupt autophagy at the phagophore formation stage, which disrupts the interaction between ATG5 and NS5B and reduces the HCV core protein to undetectable levels. Syntaxin 17, which is a receptor protein and leads to autolysosome formation, decreases the release of intracellular infectious viral particles [131]. SCOTIN, IFN-β-inducible protein, may transport viral NS5A to autophagosomes for degradation. SCOTIN’s overexpression inhibits HCV replication [117,132].

MERS coronavirus infection promotes Beclin1 proteasomal degradation by SKP2 activation. Inhibition (phosphorylation) of SKP2 decreases Beclin1 degradation and enhances autophagic flux and reduces viral replication. SKP2-targeting compounds, niclosamide (NIC) and valinomycin (VAL) were efficient in enhancing Beclin1 levels, LC3B-II/I ratios and autophagic flux. NIC and VAL treatment increased the number of autolysosomes, enhanced ATG14 oligomerization and reduced MERS-CoV multiplication [118].

At the beginning of the COVID-19 outbreak, chloroquine (CQ), a well-known anti-malarial drug, was highly regarded for the possible treatment of COVID-19. However, recent subsequent studies failed to show any clinical benefit from prescribing CQ [133]. It is proposed that CQ blocks virus infection by elevating endosomal pH, which is vital for the fusion of the virus with host cells, although questions have been raised about the interpretation of many of these studies, and by modifying the glycosylation of SARS-CoV-2 cellular receptors [134,135]. CQ also inhibits the maturation of some enveloped viruses. In addition, CQ seems to have anti-inflammatory effects through inhibiting autophagy by blocking autolysosome degradation, resulting in a decrease in cell-mediated immune response [136,137]. Although autophagy inhibition may prevent viruses from entering cells and/or may inhibit virus maturation and exit, it also prevents macrophages from presenting antigens and inhibits the activation of T cells’ and B cells’ adaptive immune responses. Recent studies show that CQ is not effective in treating patients with severe COVID-19. Moreover, it causes serious side effects, which may cause potential cardiac toxicity in patients with severe COVID-19 [138].

ACE2 may be a major host receptor for SARS, MERS, and SARS-CoV-2, which can be used to enter target cells, including cardiovascular and pulmonary cells [139]. ACE2 can inhibit cell death in both cardiac and pulmonary systems, which indicates that this protein can constrain apoptosis and autophagy-related signaling pathways. Bcl-2 is an anti-apoptotic protein that interacts with Beclin1 and Bax in an unphosphorylated form [140]. The induction of Bcl-2 phosphorylation and initiation of autophagy or apoptosis occurs in patients with underlying cardio-respiratory disease after dysfunction in the signaling pathway of ACE2/Mas/Ang (1-7) ((ACE)2-angiotensin-(1-7)-Mas receptor axis) and activates JNK. Some indirect evidence suggests that one of the key causes of increased mortality rates in high-risk patients with COVID-19 is the lower expression and inactivation of ACE-2/Mas/Ang in the cardiovascular system, resulting in the activation of the JNC/Bcl-2-Beclin1 or JNC/Bcl-2-Bax signaling pathways and the initiation of autophagy or apoptosis. The results showed that infection with COVID-19 or other coronaviruses has increased mortality rates because of an increase in the apoptosis and autophagy cycles by decreasing ACE2 expression [141]. Research has shown that human recombinant angiotensin-converting enzyme-2 (rhACE2) can interact with the SARS-CoV-2 S protein, resulting in the blockage of SARS-CoV-2 attachment and entry into host cells [60]. A recent study reported that rhACE2 inhibited the replication of SARS-CoV-2 by 1000-to-5000-fold [142].

Statins, conventionally used as cholesterol-lowering drugs, seem to be effective against viral infections by modulating different pathways. A study provided evidence that modulation of RhoA, Rabs and LC3, and inducing autophagosome maturation/degradation, may be underlying mechanisms for the inhibitory effects of simvastatin against influenza virus [143]. These authors showed upregulated LC3 protein lipidation and membrane localization by both simvastatin and H1N1; however, simvastatin plays a direct role in LC3 lipidation, but H1N1 may upregulate the expression of several ATGs.

An in vitro study on glioblastoma multiforme (GBM) tumor cells showed that lovastatin impaired autophagic flux and autophagosome-lysosome fusion machinery in these cells [144]. Statins also modulate the ACE-2 receptor and autophagy mechanisms in COVID-19 therapy [145]. Simvastatin has been reported to trigger autophagy by inhibiting the Rac1-mTOR pathway [146]. The potential beneficial effects of statins in obstructive airway diseases were demonstrated by induction of autophagy via upregulation of p53 [147] and effects on other pathways, such as the UPR [148]. Simvastatin also ameliorated inflammation, as a key asthmatic symptom, in the lungs via autophagy augmentation [149]. It was proposed that statin usage can potentially protect SARS-CoV-2-induced tissue damage through mediating ACE2 expression as one of the effective pathways in patients with COVID-19 infection. A retrospective cohort study conducted in Iranian COVID-19 patients highlighted that statins (40 mg daily) might reduce mortality in COVID-19 patients and improve lung function. They showed that using statins long-term before exposure to infection might be most effective therapeutically by controlling cytokine storm, targeting the autophagy pathway, mediating ACE2 expression, and decreasing lung scar formation in COVID-19 patients [150].

Several treatments for COVID-19 infection are being developed, including those based on the blockage of autophagy flux directly, using rapamycin or mTOR activators, or indirectly by pharmacologic agents like chlorpromazine, which produces incomplete autophagy and inhibits clathrin-mediated endocytosis. A drug named Ivermectin is a specific inhibitor of nuclear import and an in vitro SARS-CoV-2 replication inhibitor that induces autophagy by the AKT/mTOR signaling pathway [151,152]. Melatonin reduces oxidative stress, reinforces UPR, and unlocks the autophagy blockage caused by the virus, allowing autophagosome maturation and a decrease in double-membrane vesicle formation and viral replication capacity [55]. Moreover, photothermal therapy could enhance ROS production, which is shown to downregulate the Akt-mTOR-p70S6K pathway, resulting in autophagy [153].

These studies suggest that specific autophagy pathway inhibitors could become important components of drug combination therapies to combat SARS-CoV-2 infection, and highlight the potential ability of autophagy inhibitors to improve the treatment options for COVID-19 [154].

### 3.3. Role of Autophagy in the Innate Immune System

Autophagy can regulate the innate and adaptive immune responses and act as a negative modulator of inflammation through different mechanisms such as removing inflammasome-activating stimuli such as PAMP from the cytosol [155,156], regulating the production of IL-β [157] and preventing cytosolic mitochondrial DNA (mtDNA) and ROS accumulation through mitophagy [158].

Macrophages, dendritic cells, natural killer (NK) cells and T cells release different cytokines such as TNF-α, IL-1β, IL-6, IL-8 and IL-10 in response to PAMP recognition by PRR [159]. Autophagy can influence the release of cytokines, and some cytokines and immune-related cells affect the function of autophagy in return. Transforming growth factor (TGF)-β, IFN-γ, IL-1, IL-2, and IL-12 induce autophagy and IL-4, IL-10, and IL-13 inhibit autophagy [160].

Innate immunity can activate autophagy through TLRs and NOD-like receptors (NLRs) by NK T cell activation, phagocytosis, and cytokine production [161]. TLRs activate TRIF/RIP1/p38MAPK, JNK and ERK signaling pathways, or in a manner dependent on MyD88 to trigger autophagy. NLRs directly increase autophagy through recruiting and interacting with ATG16L1 [161].

Interferon regulatory factor 8 (IRF8) is an effective regulator for autophagy maturation and innate immune responses by directly promoting autophagosome formation and lysosomal fusion [162]. The exacerbated production of cytokines (cytokine storm) and chemokines along with the absent, or delayed, response of IFN-I and IFN–III lead to severe COVID-19 infection [163,164,165]. Therefore, dysregulation of inflammatory processes is the main feature of severe COVID-19 and controlling excessive production of immunological mediators should be considered as a possible therapeutic intervention [106].

## 4. Discussion: Autophagy/UPR/mTOR/NRP1 Cross-Talk in SARS-CoV-2 Contagiousness as a Possible Target for Antiviral Activities and Innate/Acquired Immunity Modulation

Generally, there is cross-talk between signaling pathways such as UPR, mTOR, autophagy, and apoptosis. The mTOR and UPR pathways regulate cellular processes, including inflammation, energy metabolism, autophagy, and apoptosis. Various cellular threats, such as nutrition/oxygen starvation, or pathogen exposure, induce ER stress, leading to the activation of the UPR pathway. Continued ER stress activates autophagy, and the UPR can trigger apoptosis during prolonged ER stress [166]. Hence, it seems that SARS-CoV-2 infection is related to the above signaling pathways in host cells.

So far, recent studies have described the role of two cell surface receptors, ACE2 and neuropilin (NRP), in virus entry into host cells and its subsequent replication, which can explain why SARS-CoV-2 is more contagious than other SARS-CoVs [12]. One reason is the presence of a sequence containing a polybasic motif, Arg-Arg-Ala-Arg (RRAR), in the SARS-CoV-2 spike protein and lack of this sequence in other CoVs. This motif increases the pathogenicity of SARS-CoV-2 by creating potential binding sites to cell surface receptors. In fact, proteolytic cleavage by furin exposes a conserved RRAR on spike protein, which conforms to a C-end rule (CendR) motif that binds to the NRP1 and NRP2 receptors. Eventually, NRP1-mediated endocytosis of the CendR peptide of SARS-CoV-2 promotes virus entry, resulting in infection [12,167]. Therefore, furin inhibitors could be promising antiviral agents to reduce virus spread and treat SARS-CoV-2 infection.

Moreover, CendR peptide bound to NRP1 initiates macropinocytosis that is mechanistically discrete from the identified endocytosis pathways. Interestingly, both CendR cargo uptake and its intercellular transport are regulated by transcription factors. The transcription factor activity is also regulated by the mTOR signaling pathway as an autophagy inhibitory element. Indeed, it could explain the inverse relationship between CendR endocytosis and NRP1 expression with mTOR activity. When cells are treated with mTOR inhibitors such as rapamycin, CendR endocytosis will increase. However, a chemical mTOR activator is able to reduce the CendR uptake during amino acid/glucose deprivation [168].

NRPs are included in many biological processes, such as the development of neurons, angiogenesis, and immune regulation [169]. They have been expressed subsequently on several immune cell subsets, including DCs, conventional T cells, and regulatory T cells (Treg) [170]. The first report of a possible role for NRP1 in the immune system showed homophilic interactions between NRP1 on mature DCs and human T cells in primary immune response initiation, following antigen exposure [171].

There are two aspects of NRP innate response regulation; one explored role is innate DC/T-cell interactions, the other one is the specific immune regulatory effects on virus-induced IFN-alpha production of DC and its consequences. To support the immunoregulatory role for NRP1 on DC, researchers analyzed whether anti-NRP1 antibody could influence DC function. They observed at least two-fold lower virus-induced IFN-alpha production in anti-NRP1-treated DCs compared to untreated DCs [172]. Collectively, NRP1 increases the susceptibility of DCs to SARS-CoV-2 infection by mediating virus internalization to non-infected cells, which is then followed by the production and secretion of cytokines, which leads to cytokine storm and higher rates of mortality [170]. Although secreted IFN-α can activate T cell responses and remove virus-infected cells, in a dangerous condition called cytokine storm, it can hyperactivate and finally deplete T cells by activating chronic immune responses, and contribute to the progression of viral infection [169]. A recent in silico study revealed that the inhibitor activity of several natural products and drugs such as Hesperidin and Ravidasvir could inhibit the binding between virus spike protein and NRP-1. Therefore, these drugs may emerge as potential NRP1 inhibitors and might be useful in the treatment of cytokine storm conditions related to COVID-19 infection [173].

Since the mTOR pathway regulates NRP1 expression, it is conceivable that *NRP1* downregulation followed by mTOR inhibition could prevent aberrant activation of immune cells and uncontrolled cytokine production in the presence of SARS-CoV-2 infection, indicating a requirement of NRP1 in the homeostasis of the immune response [170]. In contrast, NRP1 is expressed on Tregs and exerts an immunosuppressive effect [170]. Therefore, NRP1 expression on regulatory T cells may play a central role in suppressing the cytokine storm.

Since the number of Treg cells and NRP1 expression are regulated by age and gender, patients with differential immune responses report fluctuation symptoms associated with COVID-19 infection [174]. Moreover, racial variation in T-cell function has also been described. For example, higher rates of malignancy found in African-Americans compared to their white counterparts could be explained by higher levels of T helper cells and lower levels of Treg cells—the same variation could associate with higher mortality from a cytokine storm in patients with COVID-19 [169].

Evidence has shown that the second route of viral entry, mediated by NRP1, may play a more dominant role in the underlying pathophysiological mechanism of olfactory dysfunction (OD) in patients with COVID-19 than ACE2-mediated entry. Cantuti-Castelvetri and colleagues demonstrated the abundant expression of NRP1 in almost all olfactory cells, including neuronal progenitor cells and olfactory sensory neurons (OSNs) in post-mortem olfactory epithelial specimens. Therefore, binding to NRP1 could facilitate direct entry and damage to OSNs, causing loss of smell [12].

Although the most common symptoms are related to respiratory problems, ACE2 plays a central role in intestinal infection by SARS-CoV-2, which leads to gastrointestinal (GI) tract complications such as inflammation and diarrhea [175]. This is likely because of a putative ACE2 present in gut cells, which is involved in the uptake of amino acids such as tryptophan, leading to direct activation of mTOR in the presence of nutrients [176]. The mTOR also regulates the intestinal microbiota via affecting antimicrobial peptide expression in the Paneth cells of the small bowel. Studies suggest that a tryptophan reduction followed by a partial or complete blockage of ACE2 by SARS-CoV-2, could cause a decrease in mTOR activation and intestinal dysbiosis, leading to intestinal inflammation and diarrhea [176]. This inflammation increases enterocytic intercellular spaces and intestinal permeability, allowing enhanced uptake of viral and bacterial antigens. Furthermore, a massive inflammatory response (cytokine storm) by small bowel lymphocytes, DCs, and macrophages may initiate the septic state in COVID-19 patients. Indeed, intestinal inflammation and alteration of the gut microbiota result in a more severe systemic inflammation and an imbalance of the intestine’s innate immune system [175]. Conceptually, *ACE2* downregulation caused by the internalization of SARS-CoV-2 could lead to increased activation of autophagy, which is a step followed by *mTOR* downregulation. Furthermore, autophagic degradation of the abundant intestinal epithelial brush-border Na^+^/H^+^ exchanger proteins located in enterocytes is associated with NaCl absorption. In addition, autophagy regulates Paneth cells by preventing the binding of microorganisms to the gut epithelial surface via ATG5, a required protein for autophagosome formation. Therefore, the autophagy process may interfere with gut microbiota through the ACE2/mTOR/autophagy pathway during COVID-19 infection, which leads to diarrhea [176].

Therefore, it is reasonable to hypothesize that the induction of mTOR, the central regulator of autophagy, might be an effective therapeutic avenue against COVID-19-mediated diarrhea, and its complications. On the other hand, evidence shows that SARS-CoV-2 infection induces ER stress in infected cells, thereby increasing oxidative stress and triggering cellular response pathways, such as the IRE1 and PERK arms of the UPR. The excessive oxidative stress and unfolded and misfolded proteins can induce autophagy followed by direct or indirect downregulation of *mTOR*, one of the crucial metabolism regulators of mammalian cells. Therefore, it can be concluded that SARS-CoV-2 infection might induce autophagy by UPR activation in host cells.

Although autophagy destroys viruses, it raises the inflammatory response and induces antigen presentation, by which viruses take advantage to escape the immune system, replicate, and be released. Hence, several studies are trying to develop new treatments to reduce viral infection by targeting autophagy and UPR mechanisms. The signaling of intracellular pathways such as mTOR, which is directly or indirectly associated with cellular stress/UPR/autophagy/apoptosis cross-talk may be considered a possible target for treating COVID-19 infection.

Therefore, it is recommended to keep mTOR active during SARS-CoV-2 infection, which is possible because of the crucial role of the innate immune system in driving local and systemic inflammation and cytokine release in SARS-CoV-2 infection. Virus entry into host cells, followed by CendR endocytosis, triggers the innate immune response. mTOR signaling modulates the virus entry into cells via NRP1 and NRP1 expression. Notably, cytokines released from the immune cells are affected by NRP1 receptors on their surface. Thus, some mTOR activators abolish NRP1 receptor enhancement and CendR uptake of the virus, and then stimulate the immune system.

Based on the confirmed cross-talk between UPR, autophagy, and apoptosis, it can be speculated that maintaining mTOR activity inhibits autophagic degradation of misfolded and unfolded proteins. Over-accumulation of these proteins activates the apoptotic pathway and apoptotic death of the infected cells, which disrupts the virus replication cycle.

Eventually, the reduction in mTOR activity decreases the production of antimicrobial peptides produced by Paneth cells and also activates autophagy, and contributes to intestinal dysbiosis and gastrointestinal problems related to intestinal microbiota imbalance. Thus, altered microbiota might be restored by using chemical mTOR activators.

## 5. Conclusions and Future Perspectives

Accumulating evidence shows that SARS-CoV-2 infection induces ER stress in infected cells, thereby increasing oxidative stress and triggering cellular response pathways, such as the IRE1 and PERK arms of the UPR. Excessive oxidative stress and unfolded and misfolded proteins can induce autophagy. During coronavirus infection, UPR activation is caused by increased protein biosynthesis and folding during viral replication and acquisition of the ER membrane to produce DMVs. In addition, UPR and autophagy are related to each other, and the induction of UPR may potentially promote autophagy. Therefore, it can be concluded that COVID-19 infection might induce autophagy by UPR activation in host cells.

We presently lack sufficient in vivo experimental evidence concerning the role of autophagy and UPR in human viral infections, which might be related to the complications of conducting these trials. We studied the protective role of autophagy and UPR to improve the current treatment approaches. Currently, several studies are trying to develop new treatments to reduce viral infection by targeting autophagy and UPR mechanisms [177,178]. The role of autophagy in controlling viral infection has many physiological and pathological aspects. Although autophagy destroys viruses, it raises inflammatory responses and induces antigen presentation, which viruses can take advantage of to escape the immune system, replicate, and be released. Therefore, autophagy may act as a double-edged sword during viral infection. Moreover, intracellular signaling pathways such as the mTOR pathway, which is directly or indirectly associated with cellular stress/UPR/autophagy/apoptosis cross-talk, may also be considered possible targets for treating COVID-19.

However, many investigations are required to clarify the unanswered questions regarding the UPR/autophagy mechanisms in viral infections. Evidence shows that SARS-CoV-2 can enter host cells through binding to cell surface receptors, including ACE2 and NRP1. Furthermore, ACE2 blockage increases autophagy through mTOR inhibition, leading to gastrointestinal complications during SARS-CoV-2 virus infection. NRP1 is also regulated by the mTOR pathway. Increased NRP1 can enhance the susceptibility of immune system DCs to SARS-CoV-2 and induce the cytokine storm, which is related to high COVID-19 mortality (Figure 5).

Recent studies have shown that inhibition of the autophagy pathway can improve the management of viral infection complications. Autophagy could be considered a double-edged sword for viral infection. In this way, the host cellular response and type of involved cells could be decision makers. Therefore, targeting autophagy mechanisms using repositioning/repurposing procedures would potentially help in finding some efficient over-the-counter drugs such as statins, which may be promising strategies to approach viral diseases. In addition, UPR inhibitors have shown promising results in animal models and some preclinical trials for the treatment of various diseases and viral infections. Eventually, signaling pathways such as mTOR, UPR, and autophagy may be considered potential therapeutic targets for COVID-19 infection. Additionally, further research on the use of various ER stress modulating drugs is essential for improving SARS-CoV-2 treatment, which may lead to the discovery of new targets for the development of effective vaccines and therapeutics against SARS-CoV-2.

## Figures and Tables

**Figure 1 ijms-22-05992-f001:**
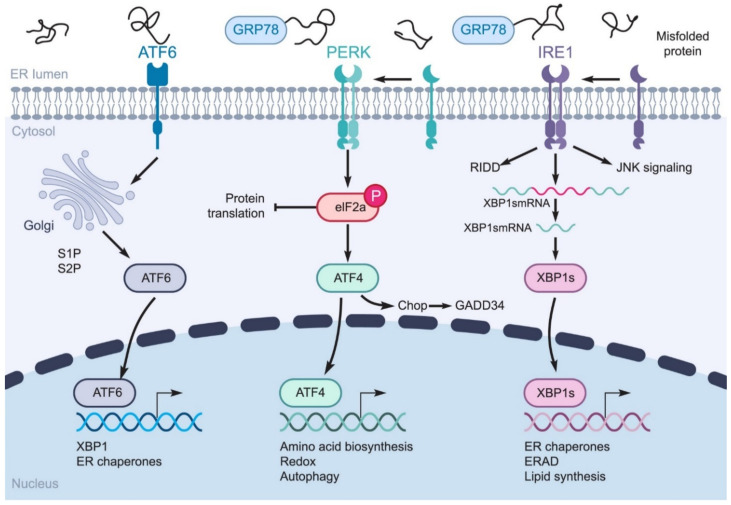
**UPR signaling pathways**. IRE1α, PERK, and ATF6 are three sensors of the UPR pathway. ER stress releases GRP78 from these sensors. After this release, activated PERK induces the eukaryotic initiation factor 2α (eIF2α) phosphorylation and ATF4 is activated and translocated to the nucleus to trigger the transcription of certain genes. IRE1α is autophosphorylated and oligomerized in response to ER stress. The activated IRE-1 provides the active form of the X-box binding protein (XBP1) transcription factor mRNA, which contributes to protein folding by regulating the expression of genes related to the proper unfolding of proteins and the secretion and destruction of the ER proteins. ATF6 is processed by Site-1 protease (S1P) and Site-2 protease (S2P). Activated ATF6 also translocates to the nucleus and, as a transcription factor, induces the transcription of ER chaperones. IRE1: inositol-requiring enzymes; PERK: protein kinase R (PKR)-like ER kinase; ATF6: activating transcription factor 6; GRP78: glucose-regulated protein 78; eIF2: eukaryotic initiation factor 2; ATF4: activating transcription factor 4; XBP1: X-box binding protein; ERAD: endoplasmic-reticulum-associated protein degradation; GADD34: growth arrest and DNA damage inducible protein 34; RIDD: regulated IRE1-dependent decay; JNK: c-Jun N-terminal kinase.

**Figure 2 ijms-22-05992-f002:**
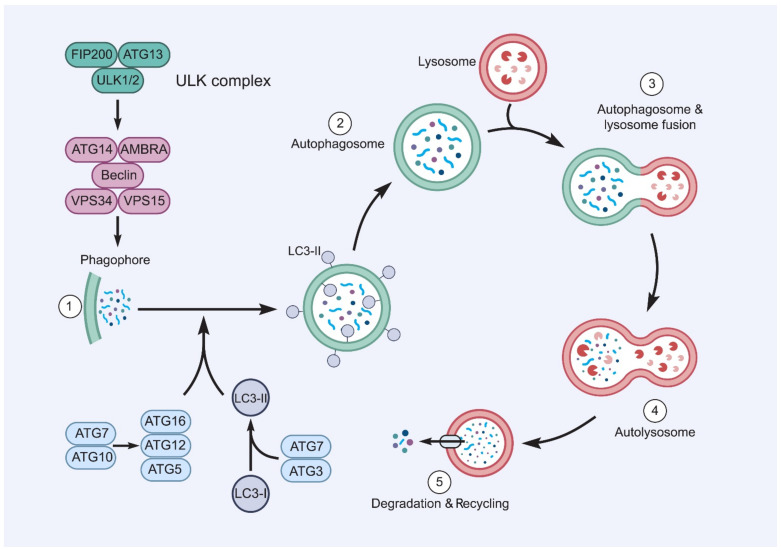
**The autophagy processes.** Autophagy is triggered by the mechanisms inactivating mTOR, and then autophagosome synthesis participates in the attachment of LC3-II to the membrane of the autophagosome and the production of vesicles that isolate cytoplasmic droplets with their double membrane. Proteolysis, which is the last step of autophagy, requires the fusion of autophagosomes with lysosomes, resulting in the release of degraded content into the cytosol. AMBRA: activating molecule in Beclin1 regulated autophagy; ATGs: autophagy-related genes; LC3-I, II: light chain I, II; VPS: vacuolar protein sorting; ULK1: unc-51-like kinase 1; FIP200: FAK family kinase-interacting protein of 200 kDa.

**Figure 3 ijms-22-05992-f003:**
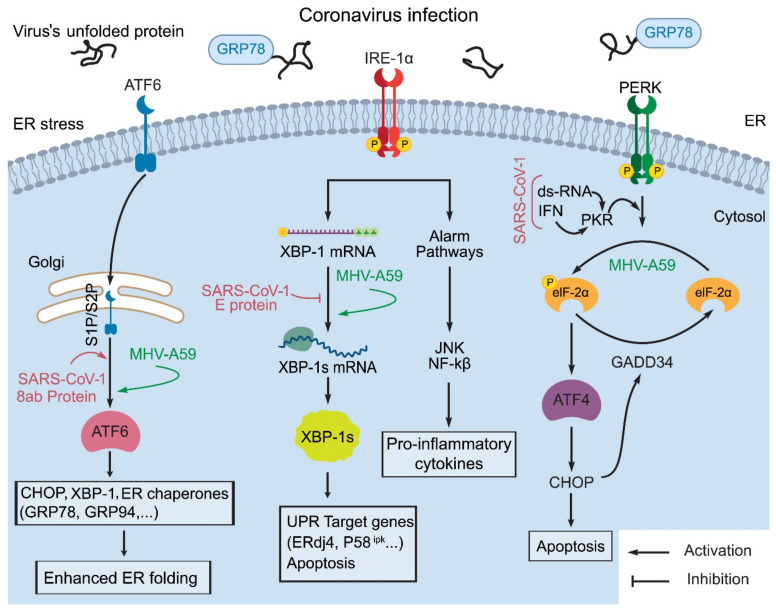
**Induction of three ER stress sensors during coronavirus infection.** This activation can affect coronavirus folding, transportation, and degradation as well as apoptosis induction in prolonged circumstances. The SARS-1 and MHV-A59 infections mainly trigger the phosphorylation of eIF2α, mostly by PKR and PERK activation. The IRE1 activation mediates XBP-1 splicing and consequently induces UPR genes such as *ERdj4* and *p58^IPK^* (58 kDa inhibitor protein kinase), which is inhibited by the SARS-1 E protein. Finally, ATF6 is processed by Site-1 protease (S1P) and Site-2 protease (S2P). ATF6 activation also increases the folding capacity of the ER, which has been verified in the MHV-A59 infection model, and the transfection of SARS-1 8ab protein. eIF2: eukaryotic initiation factor 2; PERK: protein kinase R (PKR)-like ER kinase; IRE1: inositol-requiring enzymes; XBP1: X-box binding protein; ATF6: activating transcription factor 6; GRP78: glucose-regulated protein 78; ATF4: activating transcription factor 4; *ERdj4*: ER-localized DnaJ 4; *p58^IPK^*: 58 kDa inhibitor protein kinase; S1P/S2P: Site-1 protease/Site-2 protease; IFN: interferon; GADD34: growth arrest and DNA damage inducible protein 34.

**Figure 4 ijms-22-05992-f004:**
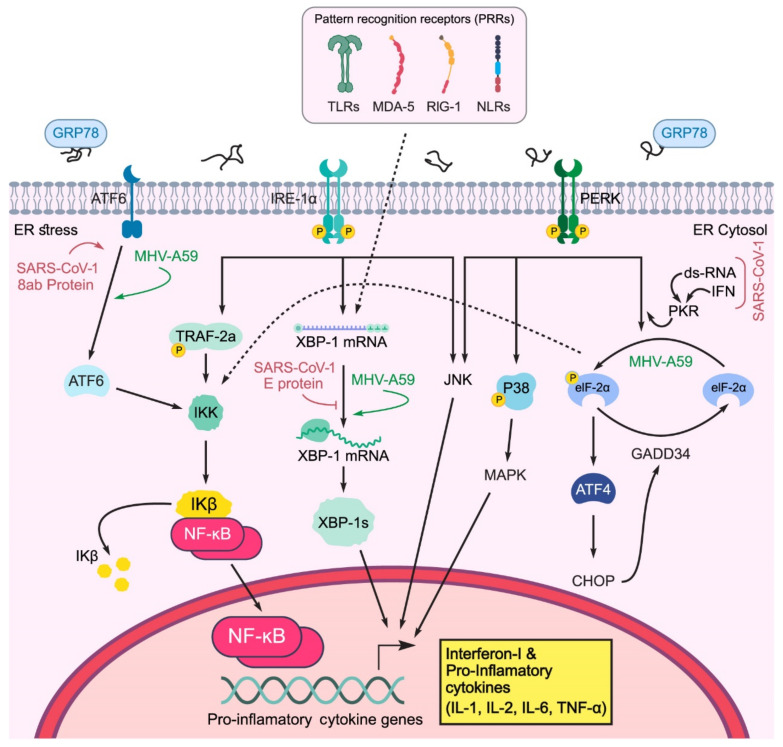
**The impact of coronavirus infection on innate immune responses**. Phosphorylation of eIF2α is triggered by the PKR/PERK pathway and leads to reduced IκB synthesis. Activated IRE1 binds to TRAF2 and activates NF-κB, which causes the production of INFI and pro-inflammatory cytokines. Activated JNK and MAP kinases p38 at the downstream of ER stress responses also up-regulate cytokine production. eIF2: eukaryotic initiation factor 2; PERK: protein kinase R (PKR)-like ER kinase; IRE1: inositol-requiring enzymes; TRAF2: TNF receptor-associated factor 2; XBP1: X-box binding protein; ATF6, 4: activating transcription factor 6, 4; GRP78: glucose-regulated protein 78; IFN: interferon; TLR: toll-like receptor; RIG-1: retinoic acid-inducible gene-I-like receptor; NLR: nucleotide-binding oligomerization domain (NOD)-like receptor; GADD34: growth arrest and DNA damage inducible protein 34.

**Figure 5 ijms-22-05992-f005:**
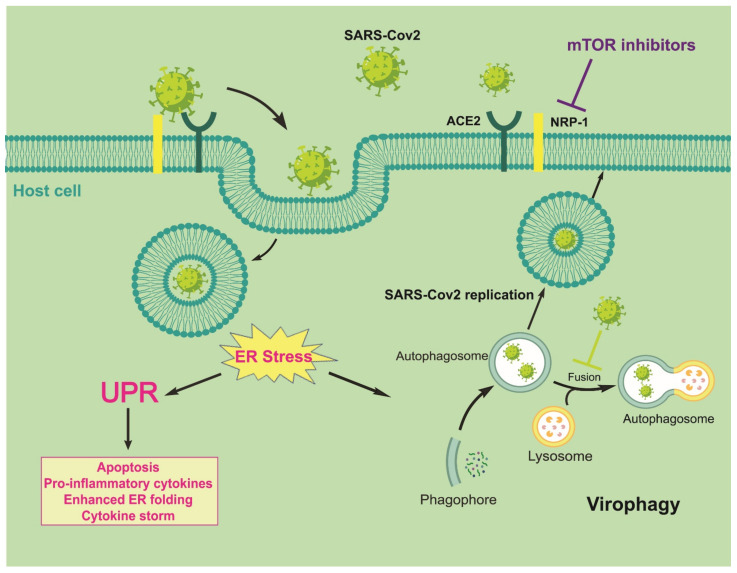
Possible cross-talk between virophagy (When the pathogen is a virus, degradation of viral components is called virophagy, which is one type of selective autophagy), UPR, and neuropilin-1 in SARS-CoV-2 infection.

**Table 1 ijms-22-05992-t001:** Summary of studies evaluating ER stress responses in selected pathogenic coronaviruses which display more similarities to SARS-CoV-2.

CoV Strain	Host System			Main Finding	Author & Year
MERS-CoV and bat coronavirus HKU9	Cells	A549		MERS-CoV and bat coronavirus HKU9 can use GRP78 as an extra target for attachment, which can exemplify the evolutionary adaptation of animal coronaviruses to human	Chu H, 2018 [56]
		AD293			
		HeLa			
		Huh7			
		Caco2			
		VeroE6			
	Human	Primary human monocyte-derived macrophages			
		Primary T cells were isolated from PBMCs			
SARS-1	293/ACE2 cells			Infection with SARS-CoV leads to the activation of PKR/PERK and triggers apoptosis independent of eIF2α phosphorylation	Krähling V, 2009 [44]
SARS-1	Cells	COS-1		The SARS-CoV 3a protein induces ER stress responses and degrades type 1 interferon receptor, which may reduce antiviral immune responses	Minakshi R, 2009 [65]
		Vero			
		Huh7			
MHV-A59	17Cl-1 cells			All arms of UPR are up-regulated with MHV-A59 infection	Cook GM, 2019 [45]
MERS-CoV	Cells	HeLa-R19		The MERS-CoV 4a protein, as an effective stress antagonist, suppresses PKR-mediated stress responses that may promote apoptosis	Rabouw HH, 2016 [66]
		Huh7			
		BHK-21			
		Vero			
SARS-1, MHV-A59	L-ACE2 cells			In CoV-infected cells, ER stress responses are induced by up-regulating XBP-1 mRNA and Herpud1, resulting in cytokine overexpression	Versteeg GA, 2007 [46]
MHV-A59	Cells	DBT		Coronavirus infection by inducing UPR, suppresses the host cell protein synthesis to accelerate the viral protein translation	Bechill J, 2008 [47]
		HeLa-MHVR			
SARS-1	Cells	HeLa		The SARS-1 8ab protein stimulates UPR to enable viral protein folding and processing by attaching to the luminal domain of ATF6	Sung S-C, 2009 [49]
		Vero E6			
SARS-1	Cells	Animal	Vero E6	The envelope protein of SARS-1 enhances ER stress responses and pro-inflammatory cytokine expression	DeDiego ML, 2011 [48]
			MA-104		
			FRhK-4		
			PK15		
		Human	CaCo-2		
			Huh7		
			HepG2		
			293		
			293T		

## Data Availability

Not applicable.

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
