# Peer review of "Autophagy, Unfolded Protein Response, and Neuropilin-1 Cross-Talk in SARS-CoV-2 Infection: What Can Be Learned from Other Coronaviruses"

_ijms, 2021, doi:10.3390/ijms22115992_

Round 1

Reviewer 1 Report

In this manuscript, Siri and colleagues have gathered current knowledge regarding the crosstalk between autophagy, unfolded protein response and neuropilin-1 in response to SARS-Cov-2 infection. They also draw some parallel with other coronaviruses. This would be of potential interest concerning the identification of compounds able to act as antiviral molecules.

The manuscript is scientifically sound and adequately illustrated. It should be of interest to scientists working in the field of UPR and autophagy and their relation to coronavirus infection.

Reviewer 2 Report

Siri et al., summarized recent works regarding SARS-CoV-2 infection in association with autophagy, unfolded protein responses  (UPR), innate immunity, and neuropilin-1 receptor. The authors further explored the discussion by considering the known networks between the key proteins involved in the autophagy, UPR and innate immunity. The authors concluded that the modulation of the key proteins involved in ER stress and autophagy may leads to the development of drugs to SARS-CoV-2.

This revised manuscript is remarkably improved and will contribute to comprehensive understandings of molecular networks of autophagy proteins during SARS2 infection.

This manuscript is a resubmission of an earlier submission. The following is a list of the peer review reports and author responses from that submission.

Round 1

Reviewer 1 Report

The review paper by Siri at el. with the title “Autophagy, UPR, and neuropilin-1 cross-talk in SARS-CoV-2 infection: A possible targeting for antiviral activities” reviews the hot topic of ongoing pandemic by COVID-19 and possible future treatment by modulating autophagy and UPR. However, the review is hard to read and follow, it is quite disperse, and in some paragraphs repetitive. The message of the review is not very clear. Also, some conclusions are exaggerated or there is no enough evidence for it. They do not clearly mention lack of studies on COVID-19, UPR, and autophagy, but rather make assumptions based on studies on other coronaviruses, clear evidence of this is Table 1, where there is no study mentioned on SARS-CoV-2. In addition, they do not clearly explain why virus hijacks autophagy and in the Figure 4 they should include in which steps of autophagy they propose the possible treatments for COVID-19. They did not include studies where some ATG proteins where KO and how this affects virus infection and replication.   

Moreover, there are many grammar mistakes (native speaker should corrected the manuscript) and in some cases, the words are used incorrectly, as for example:

  • “establishing hemostasis” (line 167) – I guess they wanted to say homeostasis;
  • “lysed form of ATF6” (line 192) – proteins are not lysed, they are degraded, cleaved, stabilized, but lysed is never used for proteins, but rather for cells;
  • “proteasomal lysis” (line 458) – again wrong use of word.
  • The PERK, IRE1 and ATF6 are not sensors, or this word in never used, but they are rather considered as UPR branches or arms; sensors are other proteins as GRP78
  • “malignancy” line 624 – this word is normally used in cancer, not for viral infections

Minor comments:

  • Line 108 – conclusion a little bit speculative, especially since in line 100/101 they state that spike from COVID-19 has 20 fold higher affinity to ACE-2 than the one from SARS-1
  • Line 112, hunger should be replaced by starvation
  • Paragraph that starts in line 177 – how is PERK more induced?
  • Line 194 – the protein will be lost? Is lost or is it not lost
  • Line 204 – mistake in the name of the virus
  • Line 212 – SARS-CoV-2 causes oxidation of what?
  • Paragraph 3.2 – Is the in silico model proven in cells or in vivo? How much of GRP78 is actually on cell membrane and how much in comparison to ACE2 so that inhibiting it would really make a difference?
  • Sentence that starts in line 264 with “In this way,” – what does it have to do with the UPR and infection of the COVID-19?  
  • Line 273 – apoptosis in some cells – which cells?
  • Conclusion in 289/290 – I do not agree that E-deficient CoVs may be a good candidates for vaccine productions or at least they should explain better, why they think so
  • Line 319 – not infection, the infection does not rely on UPR, but probably replication and propagation of the virus
  • Paragraph starting in line 383 – vague, not clear what they want to say; which cytokines, in which tissues, doing what exactly
  • Paragraph 4.2 – speculative and I don’t see the importance of it for the review
  • Paragraph 4.4 – there are reports opposing the effectiveness of CQ treatment, and also reports of serious side effects like cardiac problems
  • Line 499 – the death signal? What exactly you mean by this, especially since before you also state apoptosis
  • Sentence that starts in line 501 is incorrect and should be rephrased
  • Paragraph about statins is speculative at least to say
  • Section 4.5 is quite confuse and hard to follow, some statements are very speculative and also wrong
  • What about inhibitors of furin as possible treatment?

Reviewer 2 Report

This manuscript by Siri and colleagues provides an up-to-date review of the literature regarding the relation between SARS-CoV-2 infection and the crosstalk between Autophagy, unfolded protein response and neuropilin-1.

The manuscript is scientifically sound and properly illustrated.

A couple of minor comments remain:

  1. Figures could be cited more extensively across the main text.
  2. Figure 4: LC3-II is present at both the inner and outer membrane of the autophagosome. Only the autophagosome should be represented as a fully closed vesicle, not the phagophore (cf other published reviews on the topic).
  3. Line 446: LC3beta -> LC3B.

Reviewer 3 Report

Siri et al., summarized recent works regarding SARS-CoV-2 infection in association with autophagy, unfolded protein responses  (UPR), innate immunity, and neuropilin-1 receptor. The authors further explored the discussion by considering the known networks between the key proteins involved in the autophagy, UPR and innate immunity. The authors concluded that the modulation of the key proteins involved in ER stress and autophagy may leads to the development of drugs to SARS-CoV-2. Although, there is a concern that the SARS-CoV-2 related papers cited in this manuscript were published in 2020-2021 and have not been fully validated, this review covers wide range of the latest works and will help the readers for understandings of these topics.

Major comments

The advantage of this review paper is the description of the relationships between UPR-ER stress, autophagy, innate immunity during SARS-CoV-2 infection through the discussion of molecular networks involved. For better understandings, consider the improvement of the figures as commented below:

Figure 1-3. explain the all the molecules shown in the figures by figure legends. For example, S1P and S2P are not mentioned in the legend.

Figure 2. Is GRP78 associated with unfolded viral protein? The role of GRP78 and GRP78-unfolded protein complex is not clear.

Figure 2. Please describe the consequences of signal transduction. For examples, what do inductions of CHOP, XBP-1, ER chaperones, or GADD34 contribute to coronavirus infections?

It is recommended to make a simple summary figure which contains the summary statements in lines 717-728.

Minor comments

Throughout the manuscript, use spelled-out only in the first appearance with abbreviation.

Line 107. It is not correct to say that ‘SARS-CoV-2 is more infectious than SARS-1’. The clear difference is that SARS-CoV-2 (seems) more transmissible and persisted. No scientific evidences are available why SARS1 did not spread worldwide and persisted, though SARS1 spread immediately in the beginning of emergence.

Line 437. Consider the citation of the following papers.

Antiviral defense of autophagy to coronavirus:

- Guo et al., 2016. Autophagy negatively regulates transmissible gastroenteritis virus replication. Sci. Rep. 6:23864,

- Ko S, Gu MJ, Kim CG, Kye YC, Lim Y, et al. 2017. Rapamycin-induced autophagy restricts porcine epidemic diarrhea virus infectivity in porcine intestinal epithelial cells. Antiviral Res. 146:86–95

Beneficial role of autophagy to coronavirus:

- Guo et al. 2017. Porcine epidemic diarrhea virus induces autophagy to benefit its replication. Viruses 9(3):E53

Autophagy molecule is not associated to coronavirus infection:

- Schneider, et al. 2012. Severe acute respiratory syndrome coronavirus replication is severely impaired by MG132 due to proteasome-independent inhibition of M-calpain. J. Virol. 86(18):10112–22

- Zhao, et al. 2007. Coronavirus replication does not require the autophagy gene ATG5. Autophagy 3(6):581–85

Line 483-486. The effect of Chloroquine for treatment of COVID-19 is not determined. Modify or moderate the sentence starting with ‘Therefore, it can be concluded that to prevent…’.

Line 552-555. Induction of IRF3 and MAVS aggregation in autophagy impaired cells do not mean the essential role of autophagy mediated-immune response for JEV replication. The original paper [Jin, R., Zhu, W., Cao, S., Chen, R., Jin, H., Liu, Y., et al. 2013 Japanese encephalitis virus activates autophagy as a viral immune evasion strategy. PLoS ONE 8:e52909. doi: 10.1371/journal.pone.0052909] described that significant inhibition of JEV RNA expression in ATG5 or Beclin1 knock-down cells, and inhibition of viral replication was observed also in Atg7 and RIG-I double knock down cells.

Line 565. Mechanistic target of rapamycin.

Line 582. Reference 12 and 121 are already published.

Line 706-707. Provide the reference for ‘Currently, several studies are trying to develop new treatments to reduce viral infection by targeting autophagy and UPR mechanisms.